# The Impact and Progression of the COVID-19 Pandemic in Bulgaria in Its First Two Years

**DOI:** 10.3390/vaccines10111901

**Published:** 2022-11-10

**Authors:** Antoni Rangachev, Georgi K. Marinov, Mladen Mladenov

**Affiliations:** 1Institute of Mathematics and Informatics, Bulgarian Academy of Sciences, 1113 Sofia, Bulgaria; 2International Center for Mathematical Sciences-Sofia, 1113 Sofia, Bulgaria; 3Department of Genetics, Stanford University, Stanford, CA 94305, USA; 4Tilburg University, Warandelaan 2, 5037 AB Tilburg, The Netherlands

**Keywords:** SARS-CoV-2, COVID-19, pandemic, Bulgaria, excess mortality, IFR

## Abstract

After initially having low levels of SARS-CoV-2 infections for much of the year, Bulgaria experienced a major epidemic surge at the end of 2020, which caused the highest recorded excess mortality in Europe, among the highest in the word (Excess Mortality Rate, or EMR ∼0.25%). Two more major waves followed in 2021, followed by another one in early 2022. In this study, we analyze the temporal and spatial patterns of excess mortality at the national and local levels and across different demographic groups in Bulgaria and compare those to the European levels. Bulgaria has continued to exhibit the previous pattern of extremely high excess mortality, as measured both by crude mortality metrics (an EMR of ∼1.05%, up to the end of March 2022) and by standardized ones—Potential Years of Life Lost (PYLL) and Aged-Standardized Years of life lost Rate (ASYR). Unlike Western Europe, the bulk of excess mortality in Bulgaria, as well as in several other countries in Eastern Europe, occurred in the second year of the pandemic, likely related to the differences in the levels of vaccination coverage between these regions. We also observe even more extreme levels of excess mortality at the regional level and in some subpopulations (e.g., total EMR values for males ≥ 2% and EMR values for males aged 40–64 ≥ 1% in certain areas). We discuss these observations in light of the estimates of infection fatality rate (IFR) and eventual population fatality rate (PFR) made early in the course of the pandemic.

## 1. Introduction

The SARS-CoV-2 virus and the COVID-19 disease [1,2,3] that it causes have triggered the most significant acute public health crisis in more than a century. SARS-CoV-2 has spread widely in most countries around the world and has been the driver of substantial excess mortality in many of them [4,5].

The pandemic took divergent trajectories in different regions of the world, initially depending on the timing of the imposition of containment measures relative to the undetected, early, and cryptic spread of the virus and later based on some combination of the relaxation of these measures, seasonal effects, the buildup/waning of population immunity, the appearance of new variants of SARS-CoV-2 that are more contagious and/or antigenically divergent, and other factors. Some countries were heavily affected early on and then experienced further major epidemic waves; others were only hard hit at later stages of the pandemic.

By the end of 2020, Bulgaria emerged as one of the countries experiencing among the highest pandemic-related excess mortality in the world, even though it was one of the early containment success stories in the course of the pandemic, largely escaping the first major wave that greatly affected many areas in Western Europe and the Americas. As a previous analysis of ours has shown [6], the EMR value for the country by 1 January 2021 stood at ∼0.25% (more than twice the official death count, due to some combination of insufficient testing, registration of COVID-19 deaths as having occurred due to other reasons, and elevated mortality from otherwise treatable other conditions due to hospital capacity being exceeded).

Subsequently, the country experienced three more major waves, in March–April 2021, in the last few months of 2021, and early in 2022. In this study, we track the development and assess the impact of the pandemic on different demographic groups and regions in Bulgaria, up to the end of March 2022, using a combination of excess mortality analyses and SARS-CoV-2 genome sequencing surveillance.

These subsequent waves have dramatically increased the excess mortality burden in the country, and, as a result, it has become the first one (among those for which overall mortality data are available) where COVID-19-related excess deaths have exceeded 1% of the total population. Furthermore, we, continuing the trend established previously [6], observe major discrepancies between the outcomes within the country. EMR values in some regions are now approaching 2%, and they have exceeded that value for males in certain areas. In addition, mortality in the working-age group, 40–64, is approaching or has even exceeded 1%, a surprising result considering the commonly assumed dramatic age skew of COVID-19-related mortality. Despite the reduced Case Fatality Ratio (CFR) associated with the newly emerged Omicron variant, at the end of 2021, considerable excess mortality, not captured by official COVID-19 death statistics, persisted in the first months of 2022. These patterns are in stark contrast to those observed in countries in Western Europe, where excess mortality was concentrated in 2020 and decreased in 2021. They are, however, shared with most other countries in Eastern Europe, although Bulgaria still exhibits the most extreme excess mortality figures. The likely explanation for this pattern is the lower vaccination rates in Eastern Europe, particularly in Bulgaria. Finally, we discuss these findings in the context of the commonly cited figures for the infection fatality rate (IFR) of COVID-19.

## 2. Methods

### 2.1. Data Sources

All-cause mortality data for European countries and for NUTS-3 (Nomenclature of Territorial Units for Statistics) regions in Bulgaria were obtained from Eurostat [7,8]. The data featured in these datasets are sex- and age-stratified, with age groups split in increments of 5 years.

Country-level population data were collected through Eurostat [9] and were further supplemented by population data from the United Nations’ UNdata Data Service [10]. We further elaborate on this topic in the subsequent section on Potential Years of Life Lost (PYLL) and Working Years of Life Lost (WYLL) estimates.

The preliminary data from the most recent population census in Bulgaria were used for the analysis at the regional level in the country [11].

Life expectancy values at different ages were obtained from three separate sources. We acquire the full life tables for Bulgaria through the country’s National Statistical Institute [12]. Abridged life tables for all European countries were obtained from the World Health Organization’s open data platform [13]. This dataset is partitioned by age, in increments of 5 years. Abridged life tables for Bulgarian regions were created using regional mortality data for 2017–2019 collected by Bulgaria’s National Statistical Institute [12], following the methodology of the ONS [14].

COVID-19-related mortality and testing data for Bulgaria were obtained from the Bulgaria’s Ministry of Health. The dataset, which covers the period from the beginning of the pandemic until March 2022, includes information about each infected individual’s age, gender, region, date of latest COVID-19 test, status (infected, recovered, hospitalized, or deceased), hospitalization start and end dates (if any), whether they were taken into intensive care, and whether they died of COVID-19.

### 2.2. Data Availability

All datasets and associated code can be found at https://github.com/Mlad-en/Bulgaria_Regional_Mortality (accessed on 1 April 2022) and https://github.com/Mlad-en/COV-BG (accessed on 1 April 2022).

### 2.3. SARS-CoV-2 Variants Analysis

Information on the prevalence of SARS-CoV-2 variants was obtained from the GISAID database [15]. Variants were aggregated as “D614/B.1.x” for all non-named B.1 and derivatives lineages (the few Beta/B.1.351, and other variant sequences were classified as “others”), as “Alpha” for all B.1.1.7 lineages, as “Delta” for all B.1.617.2 and AY.x.x derivative lineages, and as Omicron “BA.1” and “BA.2” for BA.1.x.x and BA.2.x.x derivative lineages, respectively (the analyzed period predates the appearance of highly derived BA.2 lineages from later in 2022). The fraction of each variant was calculated on a weekly basis.

### 2.4. Excess Mortality and P-Scores

To calculate excess mortality across countries as well as across Bulgarian regions, we analyze the mortality observed between week 10 of 2020 and week 13 of 2022 and compare it to expected (baseline) mortality using the historical data for the five pre-pandemic years (2015–2019). The model we used is the Karlinsky–Kobak regression model [4].
*D_t,Y_* = *α_t_* + *β* × *Y* + *ϵ*
where *D_t,Y_* is the number of deaths observed in week or month *t* in year *Y*, *β* is a linear slope across years, *α_t_* are separate intercepts (fixed effects) for each week or month, and *ϵ* ∼ N(0,*σ*^2^) is Gaussian noise. The model prediction for a year *Y*, where *Y* = 2020, 2021, or 2022, is Expected Mortality*_t_*_,*Y*_ = $\hat{\alpha }$*_t_* + $\hat{\beta }$·*Y.*

We then establish a 95% confidence interval for the expected mortality. This range is used to calculate the excess mortality ∆*_t_* for a week or a month *t* and a year *Y* as:∆*_t,Y_* = Mortality*_t,Y_* − Expected Mortality*_t,Y._*

This calculation is done both as a sex- and age-stratified metric and an aggregated total excess mortality for a year *Y*, which we denote by ∆*_Y_*. To normalize excess mortality across countries, we calculate excess mortality per total population. To do this, we use population data from Eurostat for 2020.

Set *z_Y_*: = |∆*_Y_*|*/*^p^Var[∆*_Y_*], where Var[∆*_Y_*], is computed in [4]. If *z_Y_* is significantly below 2 for a given country, we consider the excess mortality for this country to be not significantly different from zero. In the computations related to the years of life lost metrics considered in the paper, we excluded a few countries having both *z_Y_* -values significantly below 2 (typically less than 1) for each age interval and wide confidence intervals that included 0 for the excess mortality associated with each of these age intervals.

Based on the excess mortality ranges, we also compute a *P*-score value for each country/region. A *P*-score value is defined as the ratio or percentage of excess deaths over certain period relative to the expected deaths for the same period based on historical data from the years 2015–2019 (see [15]). We calculate the *P*-score for a year *Y* as follows:$$P_{Y}:=\frac{{\;\mathrm{Mortality}}_{Y}-{\;\mathrm{Expected}\;\mathrm{Mortality}}_{Y}}{{\;\mathrm{Expected}\;\mathrm{Mortality}}_{Y}}\times 100$$

To calculate a total *P*-score, we replace each term in in the right-hand side in the formula above by the corresponding summation over the three-year period considered in our analysis. We also calculate the ratio between excess mortality and official COVID-19-attributed mortality. Due to the demonstrably low testing in Bulgaria [16] and other countries, this allows us to estimate underreported COVID-19 fatalities. We also use the total positive tests per region to compute a Case Fatality Ratio (CFR), which estimates the proportion of COVID-19 fatalities among confirmed cases.

### 2.5. Potential Years of Life Lost (PYLL), Aged-Standardized Years of Life Lost Rate (ASYR), and Working Years of Life Lost (WYLL) Estimates

Potential Years of Life Lost (PYLL) is a metric that estimates the burden of disease on a given population by looking at premature mortality. It is derived as the difference between a person’s age at the time died and the expected years of life for people at that age in a given country. As such, the metric attributes more weight to people that have died at a younger age.

We compute the PYLL across countries and Bulgarian regions by taking the positive all-cause excess mortality for all ages groups (in Eurostat, they are aggregated at 5-year intervals). For the European countries considered in our paper, we use the abridged life expectancy tables by the WHO (also aggregated at 5-year intervals), and, for the Bulgarian regions, we create abridged life expectancy tables following the ONS methodology [14], to calculate a total and average PYLL value for all countries and Bulgarian regions. To be more precise, for an age interval [*x,x* + 4] and sex *s* (if no sex is specified, we assume it is for both sexes) defined by ED([*x,x* + 4],*s*), the excess deaths, and by LE([*x,x* + 4],*s*), the life expectancy. Then, the potential years of life lost are computed as
PYLL([*x,x* + 4],*s*) = ED([*x,x* + 4],*s*) × LE([*x,x* + 4],*s*).

The total PYLL is computed by summing over all age intervals. In our computations, we take into account the margin of error for each ED([*x,x* + 4],*s*).

A limitation on this approach is the upper-boundary aggregation value for the two datasets. The all-cause mortality dataset’s upper boundary is 90+, while the WHO’s abridged life tables only go up to the 85+ age bracket. To account for this, we attribute the life expectancy of the 85+ age group to the 85–89 mortality group. We have, further, excluded the 90+ mortality group from our analysis.

Finally, we standardize PYLL values across countries by dividing the total sum value by the population and normalizing it per 100,000 people:$${\mathrm{PYLL}}_{\mathrm{std}}:=\frac{{\;\mathrm{PYLL}}_{\mathrm{total}\;}}{{\;\mathrm{Total}\;\mathrm{Country}\;\mathrm{Population}}_{0-89}}\times 100,000$$

The data for country-level populations in Eurostat have a similar limitation in the upper boundary of the age distribution (a cut-off at 85+). To mitigate this limitation, we supplement the population data from Eurostat for ages 0–84 with population size data for the 85–89 age group from the UNdata Data Service.

To compare the impact of the pandemic across European populations and Bulgarian regions with different age structures, we compute the Aged-Standardized Years of life lost Rate (ASYR) [17,18]. Let ([*x,x* + 4],*s*) be an age interval for a sex *s* in a standard life expectancy table for a given population. Denote, by *P*([*x,x* + 4],*s*), the population size of ([*x,x* + 4],*s*). Define the PYLL rate for ([*x,x* + 4],*s*) as
$${\;\mathrm{PYLL}}_{\mathrm{rate}\;}\left(\left[x,x+4\right],s\right):=\frac{\mathrm{PYLL}\left(\left[x,x+4\right]\right)}{P\left(\left[x,x+4\right],s\right)}\times 100,000$$

For the 2013 European Standard Population (ESP), denote, by *W*([*x,x* + 4]*,s*), the weight of ([*x,x* + 4],*s*) in the standard population. Define
ASYR(*s*): = ^X^PYLL_rate_([*x,x* + 4],*s*) × *W*([*x,x* + 4],*s*)
where the sum is taken over all age intervals. For a given population of sex *s*, this measure is interpreted as the years of life lost per 100,000 people (of sex *s*), if the population has the same age distribution as the ESP. We do the same for the Bulgarian regions, using a standardized population for Bulgaria based on 2019 census estimates by the Bulgarian NSI. ASYR allows for comparison of the pandemic impact on EU countries and Bulgarian regions having different age distributions.

Finally, we derive total, average, and total standardized WYLL value approximations. To accomplish this, we first assume people to be in the working-age group if they are 15 to 64 years old and, thus, exclude excess mortality for all age groups over 65. To calculate the remaining years of working life, we further assume a mean age for each age group, e.g., for the age interval 60−64, we assume a mean age at 62.5 years. This would leave this group with approximately 2.5 years until retirement. The 95% CI for EMRs, *P*-scores, and the values of all years of life lost functions can be found in our GitHub repository.

### 2.6. Limitations

Each of the presented data sources and approaches to analysis have their own limitations. Below, we discuss each one in detail.

#### 2.6.1. Limitation of Excess Mortality Measures

Influenza outbreaks in the period 2015–2019 contribute to the estimation for the expected mortality for 2020–2022. Thus, the expected mortality is an estimate of the “normal” death rate in the presence of seasonal influenza and other respiratory infections. As a consequence, excess mortality metrics for the first two years of the pandemic, during which seasonal influenza and other respiratory viruses largely disappeared [19,20,21,22], are probably depressed.

#### 2.6.2. Limitations of PYLL/ASYR/WYLL

Since PYLL, ASYR, and WYLL data only take into account fatalities, these metrics do not provide information about any worsened quality of life of surviving individuals, reduced life expectancy of these individuals, or working capacity. Metrics such as Disability-Adjusted Life Years (DALY), Quality-Adjusted Life Years (QALY), and Healthy Years of Life (HALE) metrics may illuminate further the total disease burden on the European population; however, obtaining the necessary information for these measurements is not yet possible.

As mentioned before, due to data availability limitations from Eurostat in our computations of PYLLs and ASYRs, we excluded the 90+ group. Given that countries such as France, Italy, and Spain have significant excess mortality in this age group, we also present a computation of the ASYRs including the 90+ age, by assuming 4 years of life expectancy (the average life expectancy for the 90+ age group for the European population is 4.74, according to the UNdata Data Service). By linear interpolation, 4 is approximately the life expectancy for the age interval (90, 94), assuming that the average age of deaths for this interval is in the range 92.7−93 (the average age of the COVID-19 fatalities above 90 years of age is 92.7 in Czechia and 92.1 in Bulgaria, and it is likely higher in Western countries that overall have higher life expectancy).

This rough approximation gives an upper bound of how large the ASYRs can go. It leads to 5−14% and 14−22% increases in the ASYRs for the (0–89) population of Eastern and Western European countries, respectively, but it does not yield a decrease between the inequalities of the countries from the two groups or any significant change in their ranks (see Appendix A).

The WYLL measure we present has some additional limitations. The first comes from the assumption that retirement age across European countries is 65. While it is most often assumed as a standard between European countries, there is actually some variation between individual member states [23]. Furthermore, we assume that the mean age of people who have died in a given age group is the middle of the given range, e.g., for the age group 60–64, the mean age = 62.5. It may well be a fact that a majority of the fatalities are concentrated in the upper part of the age bracket. However, since we do not have data about the different causes of mortality, but rather an aggregate total, we cannot be certain that this trend will hold true for all age groups and across different countries.

## 3. Results

### 3.1. Loss of Life as a Result of the COVID-19 Pandemic

In order to evaluate the impact of the COVID-19 pandemic on different countries in Europe, we applied excess mortality analysis for the period from the start of the pandemic until the end of March 2022, following previously established methods [4,6] (see the Methods section for details). Excess mortality measures are more objective measures of pandemic impact, as officially recorded COVID-19 mortality is often not an accurate representation of reality, due to insufficient availability of testing, inaccurate reporting, and other factors, such as second-order impacts of COVID-19 infections (i.e., overwhelmed healthcare systems not being able to provide adequate treatment) leading to fatalities that would not occur under normal circumstances. Specifically, in Bulgaria, 95% of the officially confirmed COVID-19 deaths occurred in hospitals, meaning that few of those who died outside hospitals entered the official statistics. The view that most excess deaths are due to COVID-19 is supported by the observation that the trajectory of excess deaths generally closely tracks that of officially recorded COVID-19 cases and deaths. Considerable discrepancies can be observed between official statistics and excess deaths, with excess deaths exceeding official numbers by even an order of magnitude or more in multiple countries [4], underscoring the importance of analyzing excess mortality to accurately understand the real impact of the pandemic. During its first major wave in 2020, Bulgaria exhibited not only the highest excess mortality in the European Union but also one of the highest discrepancies between excess deaths and official COVID-19 deaths, with an “undercount ratio” of 2.52× [4,6].

We previously estimated that Bulgaria had lost 19,004 lives during its first major COVID-19 wave in 2020. The updated analysis up to the end of March 2022 reveals that this number has increased to 68,569 (95% CI: ±6772), compared to an official COVID-19 death count of 36,529 [24], i.e., the current undercount ratio is 1.88× (±0.18). In 2021, results from the most recent nationwide census for Bulgaria became available, which showed a decrease in the population down to 6,520,314 [11]. Accounting for this updated denominator estimate, the EMR value for Bulgaria has now exceeded 1%, standing at 1.05%, circa 31 March 2022. This is the highest value recorded in any country, for which excess mortality data are available [4].

As crude mortality measures such as the EMR and the *P*-score (the percentage increase in mortality relative to baseline) may not be optimal for comparisons between populations with different demographic structures, we also calculated two standardized measures that control for such variation and aim at measuring the years of life lost as a result of the pandemic: the Potential Years of Life Lost (PYLL) and Aged-Standardized Years of life lost Rate (ASYR; see the Methods section for details). Figure 1 shows standardized (per 100,000 population) ASYR values for European countries in the three years of the pandemic, in total, and for males and females separately. Bulgaria exhibits the highest mortality by all measures among this set of countries (ASYR values per 100,000 were 11,516, 9157, and 13,745 in total, for females, and for males, respectively), followed by Lithuania and Romania (for PYLL values, see Appendix A). Excess mortality in Eastern Europe countries is much higher than that in Western Europe, and, curiously, is concentrated in the year 2021 rather than 2020, while the opposite pattern is observed in countries severely affected early in 2020 such as Spain and Italy. This observation is likely explained by two factors. First, the pandemic in 2021 in Europe was dominated first by the Alpha [25] and then by the Delta [26] SARS-CoV-2 variants, which are known to cause more severe disease than the ancestral wild-type (WT/D614G) virus [25,27,28,29,30]. Second, COVID-19 vaccination rates in Eastern Europe have been consistently lower than those in Western Europe (for example, only 11.5% of the population in Bulgaria had received two vaccine doses by 1 July 2021, and this number only increased to 29.6% by the end of March 2022 [24]), meaning that the Alpha wave and especially the Delta wave encountered a much larger proportion of completely immunologically naive individuals in populations in Eastern Europe than in Western Europe, resulting in the observed disproportionally higher mortality in the former. Indeed, we find a strong inverse correlation between vaccination rates and excess mortality, in particular in 2021 (Pearson *R*^2^ = 0.57, *p* ≤ 0.0001 and Spearman *r* = −0.69, *p* ≤ 0.0001 for ASYR values, and Pearson *R*^2^ = 0.56, *p* ≤ 0.0001 and Spearman *r* = −0.65, *p* = 0.0001 for PYLL; Appendix A).

We estimate that each excess death in Bulgaria resulted in 11.70, 12.70, and 10.43 years of life lost overall, for males, and for females, respectively, based on the ASYR metric, and in 12.57, 12.02, and 12.51 years of life lost overall, for males, and for females, respectively, based on the PYLL metric (Appendix A).

Finally, we observe that male mortality is consistently higher than female mortality for all the countries examined, which is consistent with previous observations [31].

### 3.2. Temporal Trajectory of the Pandemic in Bulgaria

Figure 2 shows the evolution of the SARS-CoV-2 variant composition in Bulgaria based on the available genome sequencing data [32]. The first major wave, in late 2020, was driven by WT-like (i.e., with the addition of the D614G mutation [33,34,35] but otherwise without major spike protein mutations affecting antigenic properties) B.1.x lineages. The Alpha variant came to dominate in early 2021 and drove the second wave, which was then itself replaced by the Delta variant in June–July 2021. Finally, in early 2022, the Omicron BA.1 variant [36,37] displaced Delta and triggered the fourth major wave, with the Omicron BA.2 lineage [38,39] beginning the next variant-displacement cycle, at the end of the observation period.

Figure 3 shows the trajectory of the pandemic in Bulgaria, in terms of recorded clinical impacts and excess mortality. We estimate that the first wave caused ∼19,000 excess deaths (or EMR ∼0.29% of the population), and the Alpha wave had a slightly lower peak and caused ∼15,000 deaths (EMR ∼0.23%); however, the Delta wave peaked at about the same heights as Alpha but was much more prolonged (Figure 3A,B) and, thus, caused the highest number of excess deaths—∼28,000 (EMR ∼0.43). The largest number of infections were recorded during the Omicron wave (Figure 3A), but it caused the fewest excess deaths, at ∼7000 (EMR ∼0.11%). A similar pattern is observed in the evolution of case fatality rate over time, which decreased dramatically once Omicron came to dominate (Figure 3C), which is consistent with worldwide observations of lower disease severity with the BA.1 variant than with the preceding non-Omicron ones [40,41,42,43,44,45,46,47,48].

Finally, we examined the “undercount ratio” (i.e., the ratio between excess deaths and official COVID-19 deaths). Its values were highest, in the 2.5–3× range, during the first major wave, then decreased to the 1.5–2.4× range during the Alpha and Delta waves, and further decreased to ∼1.5× during Omicron (Figure 3D). The most likely, in our view, interpretation of these patterns is that the undercount ratio is dependent on the extent to which hospital systems were overwhelmed by surges of severe COVID-19 cases; thus, the Omicron wave, which caused the fewest excess deaths, was most accurately captured in the official statistics, as proportionally fewer people died outside of the hospital system, which was able to accommodate a larger share of the severe cases than in previous waves. However, even with Omicron, large unaccounted-for excess mortality still persisted, likely due to the aforementioned issues of a lack of testing and the improper recording of causes of death.

### 3.3. Regional Mortality Patterns in Bulgaria

Next, we mapped the regional patterns of excess mortality in Bulgaria (Figure 4, Figure 5 and Figure 6 and Appendix A). Previously [6], we identified a stark difference between major population centers, especially the capital Sofia, and the peripheral provinces, explained by the unfavorable demographic structure and socioeconomic characteristics of the latter (where the long-term trend has been towards depopulation, resulting in a very high median age and an attendant decline in the availability of healthcare resources). This pattern has continued in the next three waves, and, thus, Sofia (city) still exhibits the lowest excess mortality in Bulgaria (EMR = 0.67%; Figure 4A,B). In contrast, excess mortality has reached as high as 1.8% in Vidin, 1.55% in Montana, and 1.5% in Razgrad. Overall excess mortality is below 1% in only five Bulgarian regions, with the northeast and northwest regions showing the highest values.

We observe even more extreme values for sex-specific excess mortality (Figure 4C,D)—male EMR is 2.1% in Montana and 1.95% in Vidin. Female-specific excess mortality is considerably lower in all regions, with only five of them exceeding EMR = 1% (highest in Vidin, at 1.28%).

We also examined excess mortality using the *P*-score metric for each year of the pandemic (Figure 4E,F). This analysis not only confirmed the previously discussed observation of very high excess mortality centered on the year 2021 but also showed that, in most regions, excess mortality in the first quarter of 2022 has been comparable to that in 2020, despite the less severe phenotype of the Omicron variant. This observation is explained by the successful containment measures in the first half of 2020, contrasting with the very large number of infections in 2022.

We also analyzed regional excess mortality using the standardized ASYR metric (Figure 5; for the standardized PYLL metric, see Appendix A). These comparisons revealed a somewhat different picture than crude mortality comparisons—ASYR values are not lowest in Sofia (city), and, according to the ASYR metric, the northeastern provinces of Razgrad and Silistra have been more heavily affected than the northwestern provinces of Vidin and Montana. This is likely because of the more extreme age skew of the demographic structure of the latter, which is normalized for by the ASYR metric but not by crude EMR estimates. As with the EMR metrics, even more extreme values are observed than the already very high one for Bulgaria as a whole—e.g., in Razgrad, the ASYR value approaches 16,000 per 100,000 population, whereas the ASYR value for Bulgaria is 11,516.

Considerable region discrepancies are also present regarding the documenting of the pandemic and the hospital outcomes for COVID-19 patients. Unfortunately, no serological survey (of any kind, not just the anti-nucleocapside protein ones that could distinguish evidence for previous infections from vaccination with mRNA or adenoviral vaccines that target only the spike protein) has ever been carried out in Bulgaria, but it is highly likely that, towards the end of March 2022, a majority of the population has been infected by SARS-CoV-2 (given the observed excess mortality; to be discussed further below). However, the percentage of the population that has tested positive is highest in Sofia (city), at only 14.17%, and is as low as 4.57%, in the peripheral Kardzhali region (Figure 6A). Thus, testing has been highly inadequate throughout the pandemic, with most infections remaining undocumented.

The undercount ratio between the EMR and the officially documented population fatality rate (PFR) ranges from 1.48× in the Sliven region to 2.84× in Pernik (Figure 6B). The overall CFR ranges from 2.13% in Sofia (city) to ≥7% in Razgrad and Smolyan (Figure 6C). These discrepancies are, in large part, due to the inadequate testing in some of the peripheral regions in the country, which also tend to be the ones with the lowest percentage of the population that has tested positive.

Remarkably, when focusing on the CFR for hospitalized patients specifically, we find no region in which fewer than 10% of COVID-19 patients died, and in Dobrich region the number exceeds 23% (Figure 6D), underscoring the unequal and inadequate access to high-quality COVID-19 treatment across the country.

### 3.4. COVID-19-Related Working-Age Excess Mortality in Bulgaria and Europe

Finally, we mapped the regional patterns of excess mortality for working-age populations (Figure 7). We focused on the 40–64 age group subpopulation, as COVID-19-related deaths and excess mortality are low in absolute number in the younger demographics, resulting in statistically unreliable estimates at the regional level.

In total, excess deaths in the 40–64 age group in Bulgaria amount to 11,986 (95% CI: ±693). We find that EMR values for this group exceed 0.2% in all regions, even for females, and reach as high as 0.8% for females and 1.03% for males in the Silistra region (Figure 7A,B). Working-age excess mortality has been concentrated in the northeastern and southern regions of the country (Figure 7C,D).

We also applied a standardized analysis using the Working Years of Life Lost (WYLL) metric (see the Methods section for details), which largely confirmed these regional patterns (Figure 7E)—in the Silistra region, the WYLL value exceeds 2500 per 100,000 population, followed by Razgrad and Pazardzhik. We also note that a unique feature of regions such as Razgrad and Silitra is the very high female-specific WYLL, at nearly double that observed in other areas, which also doubles the normal death rate (*p*-scores nearly or exceeding 100%).

The average working years of life lost per excess death are 8.26 for Bulgaria as a whole and 8.18 and 8.87 for females and males, respectively.

Finally, we compared working-age excess mortality across European countries (Figure 8). Bulgaria stands out in this analysis, exhibiting standardized WYLL values far in excess of those in the other countries included in the comparison (≥70% higher than the next ranked country, Romania). As in the comparison of overall excess mortality, countries in Eastern Europe exhibit considerably higher working-age excess mortality than those in Western Europe, which is concentrated in the second year of the pandemic.

## 4. Discussion

In this study, we map out the patterns of COVID-19-related excess mortality in Bulgaria across time, space, and different demographic groups. Three striking observations stand out in the available data.

First, considerable discrepancies exist in the impact of the pandemic at the regional level, with peripheral areas of the country exhibiting much higher absolute excess mortality than the capital Sofia, presumably due to the better access to healthcare resources and the more favorable demographic structure in the latter and, possibly, also the less favorable health status of the population in the former. Cardiovascular disease (CVD) is a well-known risk factor for severe COVID-19 outcomes, so we examined the correlation between the CVD burden in different Bulgarian regions and excess mortality during the pandemic (Appendix A). We find a strong positive correlation (Pearson *R*^2^ = 0.4 and Spearman *r* = 0.59) for overall excess mortality and CVD burden and a weaker correlation (Pearson *R*^2^ = 0.17, Spearman *r* = 0.43) for male-specific (ages 40–64) excess mortality; these observations support such a link as one of the contributing factors (we note that we also find no such correlation for female-specific working-age excess mortality or for the standardized ASYR and PYLL metrics; this is likely because CVD disease burden manifests itself earlier in males and ASYR and PYLL place less weight on excess mortality in the very elderly, where the CVD burden is most pronounced). The other likely major contributing factor to regional discrepancies is the unequal distribution of healthcare resources, as we previously discussed in more detail [6].

Second, overall excess mortality in Bulgaria is extremely high, as it is now well in excess of 1% of the total population. This result is very important for the overall understanding of the pandemic, as it finally places the early estimates of the potential impact of the SARS-CoV-2 virus in a proper context.

Numerous estimates for SARS-CoV-2′s IFR have been published, particularly early in the pandemic. A major survey of available data [27] estimated the age-standardized IFR for Bulgaria to be 0.873% in early 2020, decreasing to 0.565% in early 2021 (likely thanks to improved treatments). An early 2020 estimate for Belgium [49] placed the overall IFR at ∼1.5%. Published early 2020 estimates for Spain were IFR = 1.2% [50] and 1.15% [51]. For Eastern Europe as a whole, an IFR value ∼1.45% has been published [52]. Other estimates include 0.6% for the early pandemic IFR in China [53], 0.5% in Switzerland, and 1.4% in Lombardy, Italy [54], during the early 2020 wave, and meta-analysis-based overall estimates of 0.68% [55] and 1–1.5% [56].

In addition, several much lower values were also published during the first year of the pandemic, such as an IFR at 0.04% [57], a global one at ∼0.15% [58], an IFR at 0.17% [59] for Santa Clara County in California, USA, and others.

The validity of these estimates can be evaluated in light of the fact that Bulgaria’s excess mortality stood at 1.05% in March 2022 and that in some regions of the country it approached 2%. This outcome is the result of a combination of the following factors. First, a majority of the population must have been infected by that point (otherwise the IFR in Bulgaria would have to exceed 2%, which is unlikely), although how many exactly have been infected is not possible to say in the absence of an anti-nucleocapside serosurvey (and even then, seroreversion would probably bias estimates downwards). Second, reinfections became an increasingly common phenomenon, first with the arrival of the Delta variant [60] and especially after the appearance of Omicron. Third, the virulence of SARS-CoV-2 prior to Omicron was increasing, with the Alpha variant being more severe than the WT and the Delta variant being even more severe than Alpha; meanwhile, the IFR estimates from 2020 and early 2021 were based on the WT virus. Finally, vaccination in Bulgaria remained very low throughout the examined period, meaning that the Delta and Omicron waves were met with a large population of immunologically naive individuals, resulting in much higher mortality than in countries with high vaccination coverage. While deeply regrettable as a public health outcome for the country, this fact allows for the observation of the potential full impact of the pandemic, after infecting most of a population with a high median age and in the absence of vaccination, a situation that has been avoided, at least for the time being, in most other countries with similar demographic structures.

Third, we also observe extremely high excess mortality in working-age populations, far higher than that in other European countries. The EMR values, in the neighborhood of 1% in males aged 40–64 that we observe for several Bulgarian regions, are around or even in excess of many of the IFR values cited above for the whole population and well in excess of most estimates for working-age demographics in particular [61]. Therefore, the potential impact of the SARS-CoV-2 virus for working-age people may well have been underestimated previously.

## 5. Conclusions

The impact of COVID-19 in Bulgaria has been particularly severe compared to other countries in Europe (and likely globally). The country has recorded the highest observed excess mortality in the course of the pandemic, exceeding 1% of the population, and considerably more than that in some subpopulations and regions. These numbers are close to or even exceeding those expected after infection of the whole population. Extremely high excess mortality is also observed in working-age populations. The especially severe impact of the pandemic in Bulgaria is likely due to a combination of unfavorable demographics, uncontrolled viral transmission, a poorly prepared hospital system, and low vaccination uptake.

## Figures and Tables

**Figure 1 vaccines-10-01901-f001:**
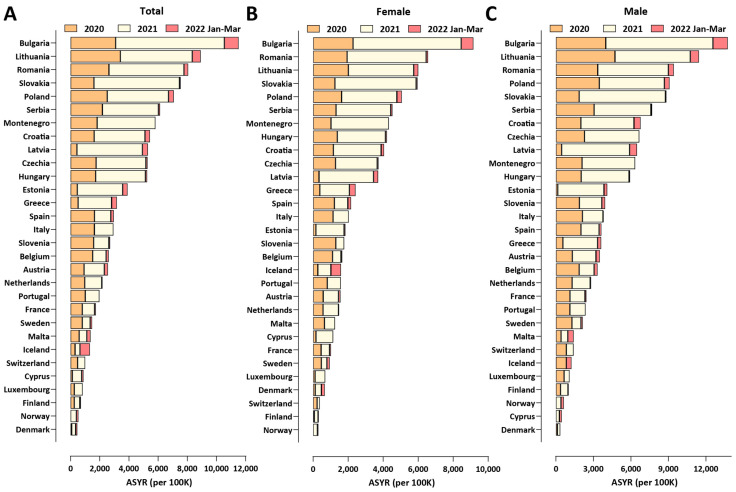
**Excess mortality in Europe and Bulgaria during the COVID-19 pandemic (up to the end of March 2022)**. (**A**) Standardized ASYR values, total; (**B**) standardized ASYR values, females; (**C**) standardized ASYR values, males.

**Figure 2 vaccines-10-01901-f002:**
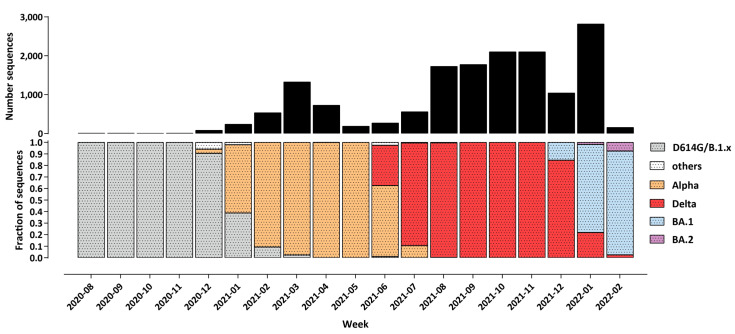
**Evolution of the SARS-CoV-2 variant composition in the course of the COVID-19 epidemic in Bulgaria up to March 2022**. Shown is the fraction of sequenced genomes belonging to each of the listed variants for each month, since the beginning of December 2020. The total number of sequenced SARS-CoV-2 genomes is shown on top.

**Figure 3 vaccines-10-01901-f003:**
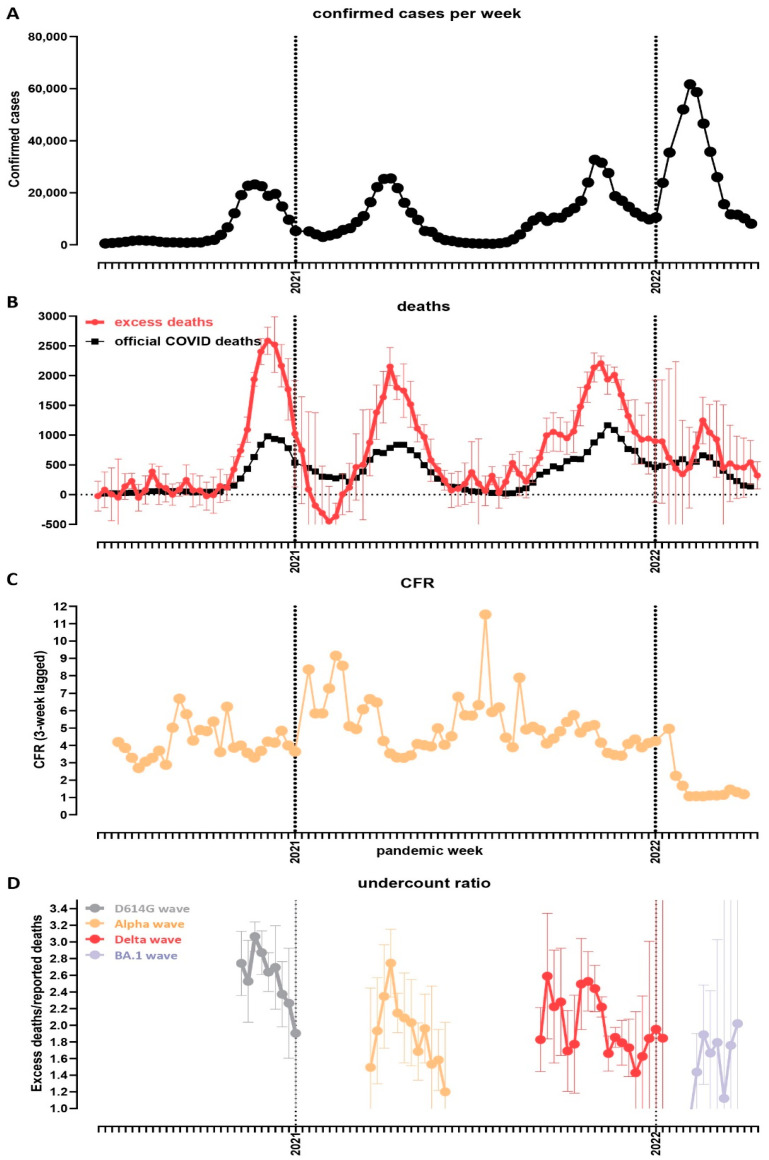
**Temporal trajectory of the COVID-19 pandemic in Bulgaria**. (**A**) Confirmed cases over time (weekly); (**B**) officially reported and excess deaths over time (weekly); (**C**) CFR over time. Data on reported cases and deaths were obtained from the Our World In Data website [24]. (**D**) Evolution of the undercount ratio (excess mortality divided by official COVID-19 deaths) over time (note that the periods between waves, for which estimates of excess mortality are uncertain, are omitted from the graph).

**Figure 4 vaccines-10-01901-f004:**
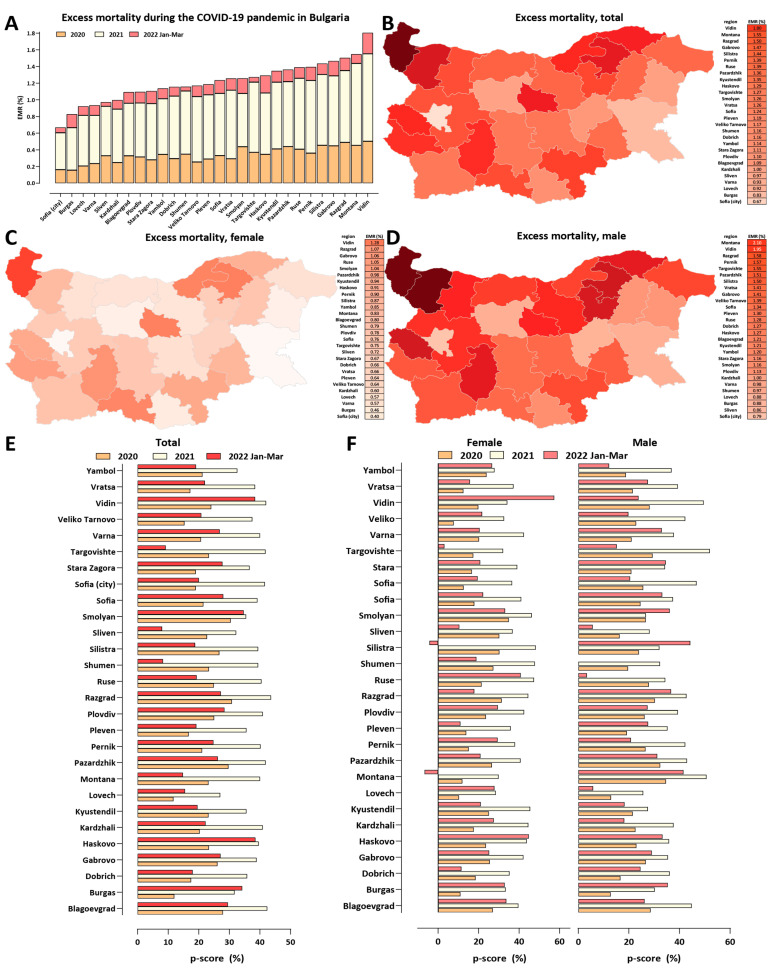
**Regional excess mortality patterns in Bulgaria during the COVID-19 pandemic (up to the end of March 2022)**. (**A**) Excess and official mortality by region and year; (**B**) excess mortality by region, in total; (**C**) excess mortality by region, for males; (**D**) Bulgaria, excess mortality by region, for females; (**E**) excess mortality (*P*-scores), per year; (**F**) excess mortality (*P*-scores), for males and females, per year.

**Figure 5 vaccines-10-01901-f005:**
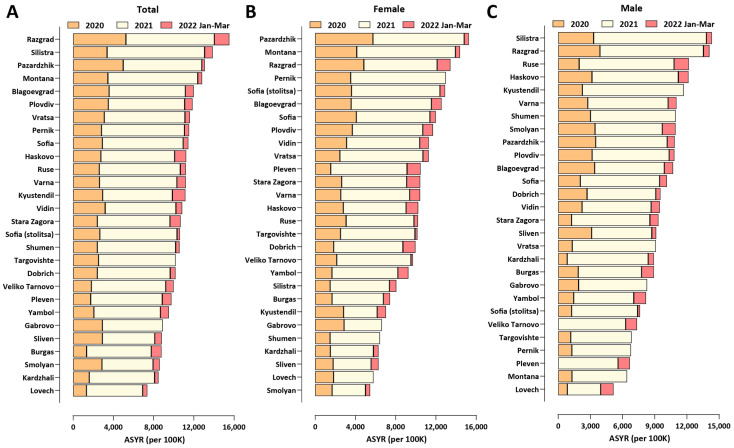
Regional excess mortality patterns in Bulgaria during the COVID-19 pandemic (up to the end of March 2022). (**A**) Total, standardized ASYR values; (**B**) Female, standardized ASYR values; (**C**) Male, standardized ASYR values.

**Figure 6 vaccines-10-01901-f006:**
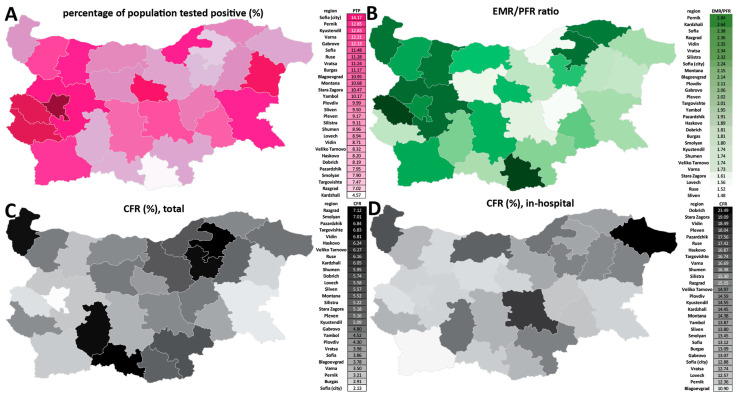
**Regional discrepancies in the extent of recording of the impact of the COVID-19 and hospital outcomes in Bulgaria**. (**A**) Percentage of the population that has tested positive for SARS-CoV-2 in Bulgarian regions; (**B**) undercount ratio for Bulgarian regions; (**C**) total CFR values for Bulgarian regions; (**D**) in-hospital CFR values for Bulgarian regions.

**Figure 7 vaccines-10-01901-f007:**
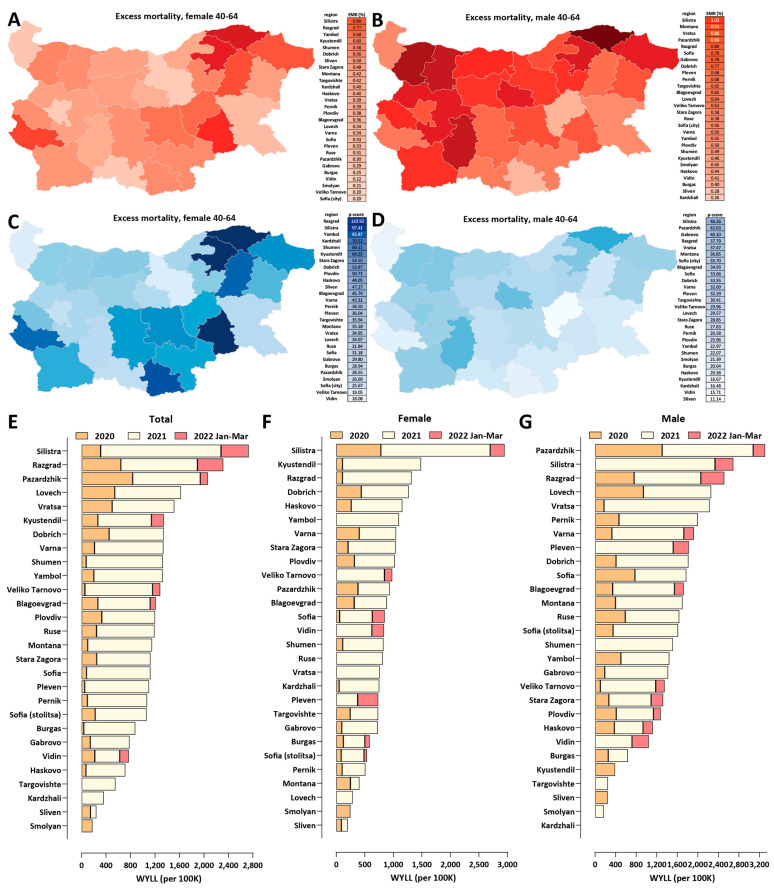
**Working-age excess mortality in Bulgaria during the COVID-19 pandemic (up to the end of March 2022)**. (**A**) Excess mortality by region, for females ages 40–64, EMR values; (**B**) excess mortality by region, for males ages 40–64, EMR values; (**C**) excess mortality by region, for females ages 40–64, *P*-scores; (**D**) excess mortality by region, for males ages 40–64, *P*-scores; (**E**) standardized WYLL values, in total; (**F**) standardized WYLL values, for females; (**G**) standardized WYLL values, for males.

**Figure 8 vaccines-10-01901-f008:**
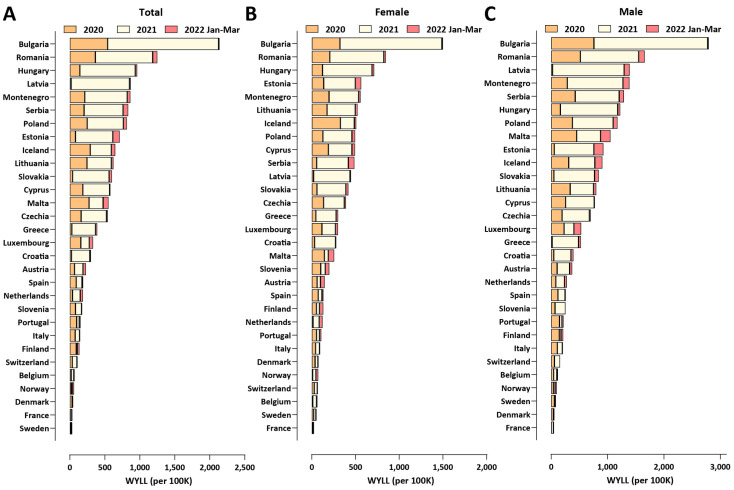
Working-age excess mortality in European countries during the COVID-19 pandemic (up to the end of March 2022). (**A**) Standardized WYLL values, in total; (**B**) standardized WYLL values, for females; (**C**) standardized WYLL values, for males.

## Data Availability

All datasets and associated code can be found at https://github.com/Mlad-en/Bulgaria_Regional_Mortality and https://github.com/Mlad-en/COV-BG (accessed on 1 April 2022).

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
