# Peer review of "The Impact and Progression of the COVID-19 Pandemic in Bulgaria in Its First Two Years"

_vaccines, 2022, doi:10.3390/vaccines10111901_

Round 1

Reviewer 1 Report

The authors analyzed the impact of SARS-CoV-2 between early 2020 and early 2022. The mortality data was retrieved from different sources and analyzed again using different models. The outcomes between Bulgaria and other European countries were compared. In addition, different regions of Bulgaria were also discussed. The manuscript was well written, the methods used were written down clearly and readers were alerted with limitations. I have some comments so that the manuscript can be improved further.

Major comments:

1. As the authors did not generate their own data, the title should be revised to avoid misleading, my suggestion: An observational study of the impact of SARS-CoV-2 in Bulgaria between 2020-2022

2.  The methods to analyze SARS-CoV-2 variants were not mentioned. Adding the reference 28 alone is not enough. The authors listed out the steps to analyze mortality data thoroughly, however, the steps for analyzing SARS-CoV-2 variants should be elaborated in parallel. If the authors think that this part is not important, they can consider to document it in supplementary section.

3. The findings of the study was appealing. The authors tried to correlate the findings with explanations, however, this part was very weak without enough evidences. The authors should list out the references one by one. Then, the claims of the manuscript would be more comprehensive and objective.

(a) varied mortality rates between regions of Bulgaria vs better healthcare resources/health status, in addition to cite the authors’ previous findings in reference 6, list out more references.

(c) less severe virulence Omicron strains when comparing with SARS-CoV-2 strains before the emerging of Omicron, the authors are required to list out the supporting evidences.

(c) vaccination in Bulgaria was low, any data to support this claim? how about the countries with low mortality rate? were they supported by high vaccination rate?

Minor comments

- The abbreviations used in the figures should list out the full terms. The figures should be read and interpreted standalone from the main text.

Author Response

Responses are provided as a separate PDF file.

Reviewer 2 Report

In this manuscript a study is reported about COVID-19 in Bulgaria, its emergence, presence and evolution starting at the start of the pandemic in 2020, and ending March 2022. The authors had to use surrogate markers like excess mortality during this period when compared with the 2015-2019 period, to estimate the consequences of COVID-19. This is because there are no data on vaccination rate (besides the fact that the vaccination rate is low compared with other countries in Europe) and no date on documented infection (also testing rate was quite low). These limitations are properly explained in the text. Despite these limitations the authors reach conclusions that are quite relevant and threefold: first, considerable discrepancies exist in the impact of the pandemic at the regional level, in particular comparing regional data with those in the capital Sofia; second, overall excess mortality in Bulgaria is extremely high when compared with other countries in east and west Europa and exceeds nowadays 1%; and third, extremely high excess mortality was observed in working age populations, far higher than that in other European countries. Although not mentioned, these conclusions deserve attention in relation to public health strategies in Bulgaria in attacking and managing COVID-19 waves in the future, because such waves are likely to occur, also in a country where many people have some immunity thanks to natural infection and no thanks to vaccination.

This is a very relevant study, not only regarding estimates for the COVID pandemic for the Bulgarian society, but also by the use of existing databases to make reliable estimates which are validated by proper tools provided by statistics. The report is written extremely well, with a clear Introduction and excellent description of data sources and methods including statistics in the Methods, a clear presentation of Results including those in the Supplementary Data, and a clear interpretation and conclusions in the Discussion. There are a few points that could be addressed in a revision:

·        Line 198: Limitation of Excess mortality measures. Question, are there influenza vaccination programs in place in Bulgaria, like in other countries? A short sentence may clarify this.

·        Line 202: Limitations of PYLL/ASYR/WYLL, Here the exclusion of the 90+ age group is discussed in relation to other Western countries.

·        Line 287-290: Indeed, we find strong inverse correlation between vaccination rates and excess mortality, in particular in 2021 (Pearson R2 = 0.57, p ≤ 0.0001, and Spearman r = −0.69, p ≤ 0.0001 for ASYR values, and Pearson R2 = 0.56, p ≤ 0.0001; Spearman r = −0.65, p = 0.0001 for PYLL; Supplementary Figure 2). Please add that these data regard European countries and not include Bulgaria, correct?

·        Following this point, the correlation is between rates of vaccination and excess mortality. This seems understandable, as there are no individual person data showing a direct correlation. It is advised to bring this in the text, as a clarification.

·        There are many abbreviations used in the report, that are not commonly used: it is advised to include a list of abbreviations.

·        The authors are advised to describe a perspective of the present study for the public health situation and strategies, like written above.

Author Response

Responses are provided as a separate PDF file
